# Pressure-induced monotonic enhancement of $T_c$ to over 30 K in superconducting $Pr_{0.82}Sr_{0.18}NiO_2$ thin films

N. N. Wang [1,2,4], M. W. Yang[1,2,4], Z. Yang[1,2,4], K. Y. Chen [1,2], H. Zhang[1,2], Q. H. Zhang[1,2], Z. H. Zhu[1,2], Y. Uwatoko [3], L. Gu [1,2], X. L. Dong [1,2], J. P. Sun [1,2] ✉, K. J. Jin [1,2] ✉ & J.-G. Cheng [1,2] ✉

The successful synthesis of superconducting infinite-layer nickelate thin films with the highest $T_c \approx 15$ K has ignited great enthusiasm for this material class as potential analogs of the high-$T_c$ cuprates. Pursuing a higher $T_c$ is always an imperative task in studying a new superconducting material system. Here we report high-quality $Pr_{0.82}Sr_{0.18}NiO_2$ thin films with $T_c^{onset} \approx 17$ K synthesized by carefully tuning the amount of $CaH_2$ in the topotactic chemical reduction and the effect of pressure on its superconducting properties by measuring electrical resistivity under various pressures in a cubic anvil cell apparatus. We find that the onset temperature of the superconductivity, $T_c^{onset}$, can be enhanced monotonically from ~17 K at ambient pressure to ~31 K at 12.1 GPa without showing signatures of saturation upon increasing pressure. This encouraging result indicates that the $T_c$ of infinite-layer nickelates superconductors still has room to go higher and it can be further boosted by applying higher pressures or strain engineering in the heterostructure films.

Since the discovery of high-$T_c$ superconductivity in cuprates[1], numerous experimental and theoretical investigations have been carried out aiming at finding more superconductors with higher $T_c$ and unveiling the mysteries mechanisms. After over 30 years of endeavor, the highest $T_c$ in cuprates, ~135 K at ambient[2] and ~164 K under pressure[3], remains to be lower than room temperature, and the mechanism is still an enigma. As the nearest neighbor to copper in the periodic table, the infinite-layer nickelates showing the great similarities in crystal structures and electronic configurations to cuprates have been considered as potential high-$T_c$ superconductors ever since the early 1990s[4-13]. Unfortunately, superconductivity was not observed in the synthesized powder and thin-film nickelates until very recently.

In 2019, Li et al. reported the experimental observation of superconductivity with $T_c$ = 9–15 K in the hole-doped infinite-layer $Nd_{1-x}Sr_xNiO_2$ thin films obtained by soft-chemistry topotactic reduction from the corresponding perovskite phase[14]. From the experimentally constructed superconducting phase diagram, the observed $T_c(x)$ has a non-monotonic evolution with doping $x$, similar to the hole-doped cuprates[15-17]. This discovery has reignited the enthusiasms on the nickelates and immediately attracted extensive investigations recently (for a review, see ref. 18). For the parent $LaNiO_2$ and $NdNiO_2$, X-ray absorption spectroscopy (XAS) and resonant inelastic X-ray scattering (RIXS) confirmed the nominal $3d^9$ electronic configuration but revealed a reduced hybridization between Ni 3d and O 2p orbitals and an enhanced coupling between Ni 3d and La/Nd 5d states[19]. Notably, the electron energy loss spectroscopy (EELS), high-resolution XAS and RIXS experiments further revealed that the doped holes reside on the Ni sites, forming the low-spin $d^8$ state[20,21]. These observations are different from those in cuprates[22,23]. Recent STM experiments uncovered a mixed s- and d-wave superconducting gap feature on the rough surface of $Nd_{1-x}Sr_xNiO_2$ thin films[24], and these results were reproduced by employing ab initio treatment on different terminated surfaces[25,26]. From the theoretical point of view, three

[1]Beijing National Laboratory for Condensed Matter Physics and Institute of Physics, Chinese Academy of Sciences, Beijing 100190, China. [2]School of Physical Sciences, University of Chinese Academy of Sciences, Beijing 100190, China. [3]Institute for Solid State Physics, University of Tokyo, Kashiwa, Chiba 277-8581, Japan. [4]These authors contributed equally: N. N. Wang, M. W. Yang, Z. Yang. ✉e-mail: jpsun@iphy.ac.cn; kjjin@iphy.ac.cn; jgcheng@iphy.ac.cn

main perspectives on nickelates have been proposed, including the cuprate-like correlated single-Ni-orbital $d_{x^2-y^2}$ band[23,25,27–32], the Ni-3d-multiorbital effects[33–37], and the Kondo physics between Ni-3d and Nd-5d orbitals[38–40]. So far, consensus has not yet been reached about the superconducting mechanism of nickelates. This is partially attributed to the great challenges in the materials' synthesis and the relatively poor quality of these infinite-layer nickelates, as well as the limited techniques for regulating their physical properties. In contrast to the superconducting thin-film samples, the bulk samples exhibit an insulating ground state, and no indication of superconductivity was observed even under pressures up to 50 GPa[41]. It has raised the question whether the observed superconductivity correlates intimately with the heterostructure or epitaxy strain between the thin films and the substrate.

In addition to $Nd_{1-x}Sr_xNiO_2$ thin films, superconductivity has also been achieved in other doped rare-earth nickelates, such as $La_{1-x}(Ca/Sr)_xNiO_2$[42,43] and $Pr_{1-x}Sr_xNiO_2$[44]. The phase diagrams of $(La/Pr/Nd)_{1-x}$ $Sr_xNiO_2$ and $La_{1-x}Ca_xNiO_2$ thin films are all featured by the superconducting phase adjacent to the weakly insulating state in the underdoped and over-doped regimes[16,17,42–44]. In addition, the doping-dependent generalized superconducting dome was found to shift upon changing the rare-earth cations. However, a local $T_c$ valley structure of the superconducting phase was not observed in the $Pr_{1-x}Sr_xNiO_2$[44], which is different from the situations in $Nd_{1-x}Sr_xNiO_2$ and $La_{1-x}(Ca/Sr)_xNiO_2$[16,17,42,43]. These comparisons highlight the important role of rare-earth cations played in addition to the lattice strain applied by the $SrTiO_3$ substrate in regulating the superconducting state of the nickelates. Despite of much recent effort, the reported highest $T_c^{onset}$ of the infinite-layer nickelates remains lower than 15 K[14,16,17,42–48]; it thus becomes an important issue to further enhance the $T_c$ of the superconducting nickelates to higher temperatures. In this regard, the application of high pressure that has been widely employed to raise $T_c$ of cuprates[3] and iron-based unconventional superconductors[49–52] should be a primary choice. To the best of our knowledge, however, this approach has not been applied to the

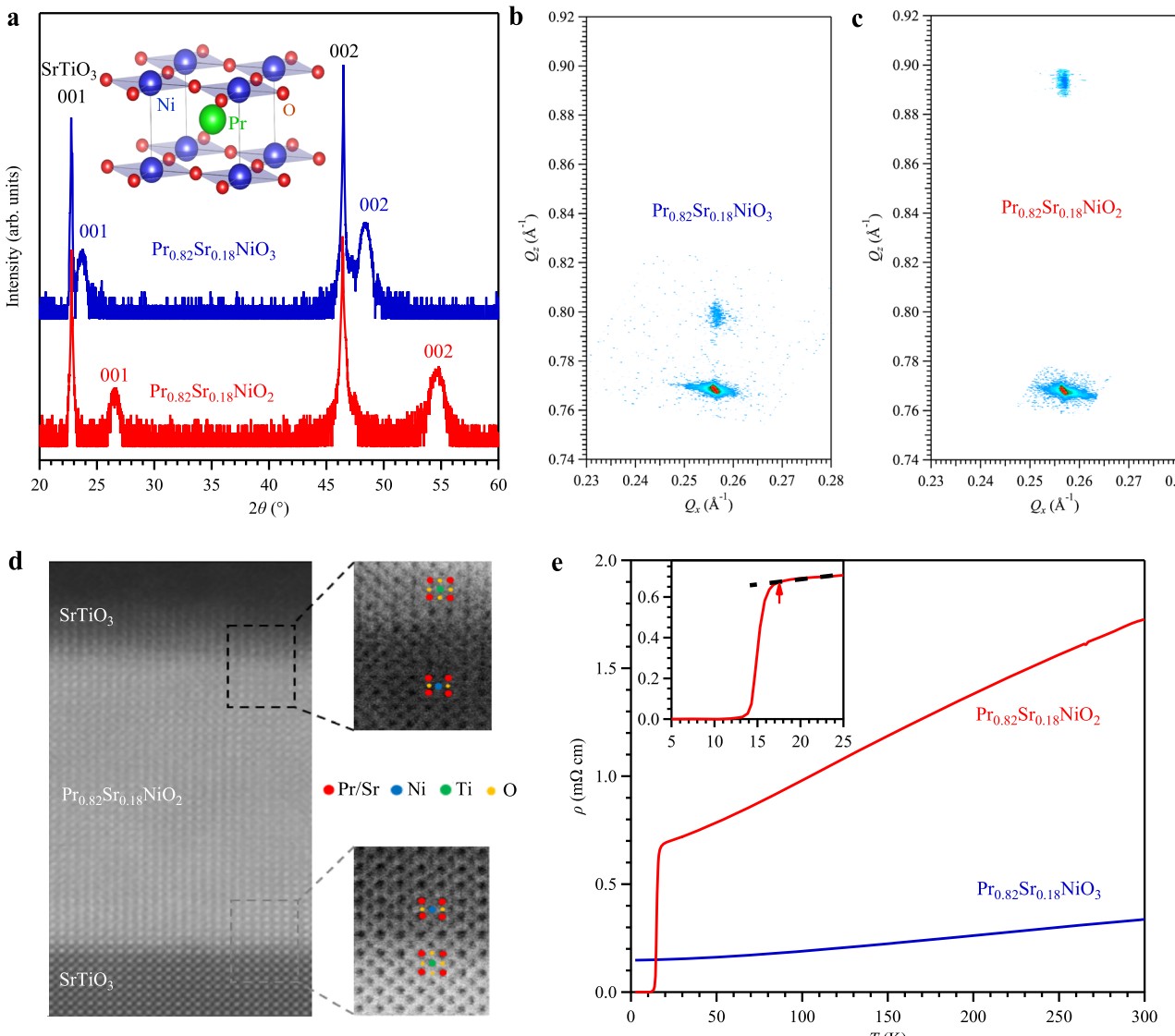

**Fig. 1 | Structural characterization and transport properties of perovskite $Pr_{0.82}Sr_{0.18}NiO_3$ and infinite-layer $Pr_{0.82}Sr_{0.18}NiO_2$ thin films. a** X-ray diffraction $\theta-2\theta$ symmetric scans of perovskite $Pr_{0.82}Sr_{0.18}NiO_3$ thin film (blue) and infinite-layer $Pr_{0.82}Sr_{0.18}NiO_2$ thin film (red). Reciprocal space mappings (RSM) of **b** perovskite $Pr_{0.82}Sr_{0.18}NiO_3$ thin film and **c** infinite-layer $Pr_{0.82}Sr_{0.18}NiO_2$ thin film, respectively. **d** The atomic-resolution HAADF-STEM imaging of infinite-layer samples in **a**. **e** Temperature-dependent resistivity for perovskite $Pr_{0.82}Sr_{0.18}NiO_3$ thin film (blue) and infinite-layer $Pr_{0.82}Sr_{0.18}NiO_2$ thin film (red) which shows a high superconducting transition temperature $T_c^{onset} \approx 17$ K, as highlighted by the red arrow in the inset of **e**. Inset of **a** shows the crystal structure of $PrNiO_2$.

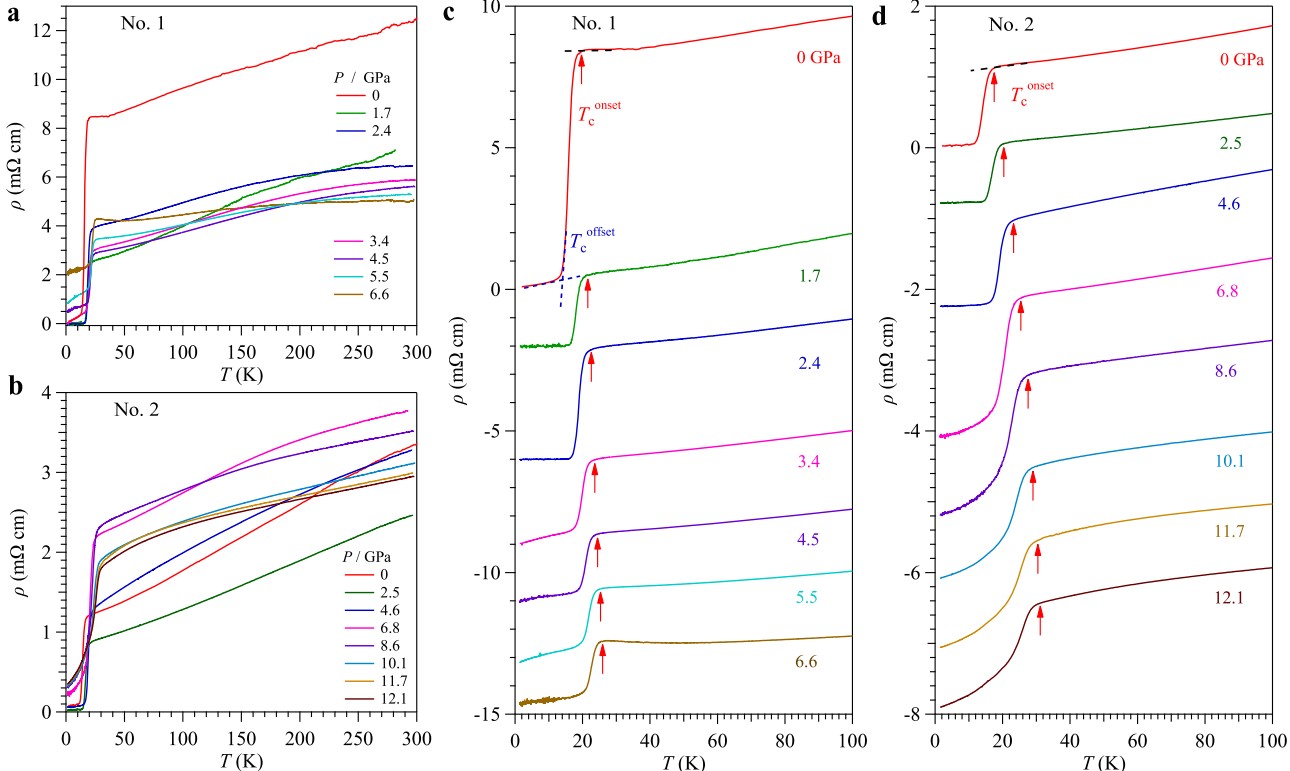

**Fig. 2 | Transport properties and superconductivity of the infinite-layer Pr$_{0.82}$Sr$_{0.18}$NiO$_2$ thin films under high-pressure.** Temperature dependence of resistivity $\rho(T)$ of Pr$_{0.82}$Sr$_{0.18}$NiO$_2$ thin films under various pressures **a** up to 6.6 GPa for sample No. 1 with Daphne 7373 and **b** up to 12.1 GPa for sample No. 2 with glycerol as PTM. The resistivity $\rho(T)$ curves below 100 K **c** for No. 1 and **d** for No. 2, illustrating the variation of the superconducting transition temperatures with pressure. Except for data at 0 GPa, all other curves in **c** and **d** have been vertically shifted for clarity. The $T_c^{onset}$ (up-pointing arrow) was determined as the temperature where resistivity starts to deviate from the extrapolated normal-state behavior and the $T_c^{offset}$ (crossing point) was defined as the interception between two straight lines below and above the superconducting transition.

superconducting nickelate thin films so far. We are thus motivated to investigate the effect of pressure on the superconducting properties of the infinite-layer nickelates.

In this work, we first synthesize high-quality Pr$_{0.82}$Sr$_{0.18}$NiO$_2$ thin films with a high $T_c^{onset} \approx 17$ K, and then perform transport measurements by using the palm-type cubic anvil cell (CAC) apparatus under various pressures up to 12.1 GPa. We observe a positive pressure effect on the superconducting transition temperature, $T_c(P)$, which increases monotonically from ~17 K at ambient pressure to ~31 K at 12.1 GPa without leveling off. This result is quite encouraging and should promote further endeavors to raise the $T_c$ of this new class of superconductors.

## Results

### Thin film preparations and characterizations at ambient pressure

The single-crystalline infinite-layer Pr$_{0.82}$Sr$_{0.18}$NiO$_2$ thin films were synthesized by two steps. First, the precursor perovskite phase Pr$_{0.82}$Sr$_{0.18}$NiO$_3$ films were deposited on TiO$_2$-terminated SrTiO$_3$ (001) substrates and then capped with the SrTiO$_3$ epitaxial layer by using the pulsed laser deposition system[53,54]. Subsequently, the thin films were vacuum-sealed with CaH$_2$ in a glass tube for the ex-situ topotactic reduction process to obtain the infinite-layer films. Details about the sample syntheses can be found in the Methods. Figure 1a shows the X-ray diffraction (XRD) $\theta$−$2\theta$ symmetric scans of thin-film samples for perovskite Pr$_{0.82}$Sr$_{0.18}$NiO$_3$ (blue) and infinite-layer Pr$_{0.82}$Sr$_{0.18}$NiO$_2$ (red), showing only (001) and (002) peaks for both phases. After reduction, the rightward shift of the (001) and (002) peaks in the infinite-layer phase corresponds to a compression of the out-of-plane lattice constant, which is calculated to be 3.36 Å. As seen from the

reciprocal space mappings in Fig. 1b, c, both films are closely fixed to the in-plane SrTiO$_3$ lattice. To further characterize the infinite-layer thin films, we performed the cross-sectional scanning transmission electron microscopy (STEM), Fig. 1d. The infinite-layer structure of the Pr$_{0.82}$Sr$_{0.18}$NiO$_2$ thin film can be intuitively seen from the atomic-resolution high-angle annular dark-field (HAADF) images, elaborating the high-quality of the synthesized samples.

Figure 1e shows the temperature-dependent resistivity $\rho(T)$ for both the perovskite Pr$_{0.82}$Sr$_{0.18}$NiO$_3$ and infinite-layer Pr$_{0.82}$Sr$_{0.18}$NiO$_2$ thin films. For Pr$_{0.82}$Sr$_{0.18}$NiO$_3$, the $\rho(T)$ shows a metallic behavior with a weak temperature dependence over the whole temperature range. Pr$_{0.82}$Sr$_{0.18}$NiO$_2$ also displays a typical metallic behavior upon cooling down and exhibits a pronounced superconducting transition below $T_c^{onset} \approx 17$ K, which is defined as the temperature where the resistivity deviates from the linear extrapolation of normal-state resistivity, inset of Fig. 1e. Zero resistance is achieved at $T_c^{zero} \approx 11$ K, which is in good agreement with previous reports[44].

### High-pressure resistivity

To further raise $T_c$ and tune the physical properties of this new superconducting family, we study the pressure effect on the Pr$_{0.82}$Sr$_{0.18}$NiO$_2$ films by measuring resistivity under various hydrostatic pressures. The experimental details can be found in the Methods. Figure 2a, b shows the $\rho(T)$ of two Pr$_{0.82}$Sr$_{0.18}$NiO$_2$ thin films (No. 1 and 2) measured under various pressures in the presence of Daphne 7373 and glycerol as the liquid pressure transmitting medium (PTM), respectively. As seen in Fig. 2a, the magnitude of normal-state resistivity shows complex and non-monotonic evolutions with increasing pressure. At $P > 1.7$ GPa, a broad hump feature appears in $\rho(T)$ between 100 and 300 K, which is similar to the situation seen in the Fe-based

superconductors due to the incoherent-to-coherent crossover of the $3d$ electrons[55–58]. When we further increase pressure to 6.6 GPa, an obvious upturn in $\rho(T)$ appears below ~50 K, and the superconducting transition emerges at $T_c^{onset} \approx 26$ K. Moreover, we can see that the residual resistivity at 1.5 K exhibits a prominent enhancement with increasing pressure above 2.4 GPa; it increases to the value that is about half of the normal-state value at 6.6 GPa. Such a large enhancement should be induced by the solidification of the Daphne 7373 at high pressures and low temperatures as discussed below. For the sample No. 2 in the glycerol PTM, the normal state $\rho(T)$ and the residual resistivity show similar evolutions with pressure as sample No. 1; it first decreases considerably from 0 to 2.5 GPa, increases gradually until 6.8 GPa, and then decreases again at higher pressures. These high-pressure results resemble the observations in the Sr/Ca-doped infinite-layer nickelate thin films[14,16,17,42–44], in which the defects or cation-site disorders dominate the scattering processes in resistivity. As discussed below, the delicate thin films are partially deteriorated upon compression, especially at higher pressures when the liquid PTM solidifies and produces more disorders and defects that enhance the carrier scatterings. In the presence of liquid PTM, the superconducting transition remains visible up to at least 12.1 GPa, the highest pressure in the present study.

To highlight the evolutions with pressure of the superconducting transition, we vertically shift the $\rho(T)$ curves below 100 K for the samples No. 1 and No. 2, as displayed in Fig. 2c, d. We can clearly see that the $T_c$ increases monotonically with pressure in the studied pressure range. In addition to $T_c^{onset}$ (red arrow, as defined above), here we also define the $T_c^{90\%Rn}$ as the temperature where the resistivity drops to 90% of $\rho(T_c^{onset})$ [$T_c^{onset}$ and $T_c^{90\%Rn}$ correspond to the temperatures where the superconducting state (or Cooper pairs) just appears and grows up quickly], and $T_c^{offset}$ as shown in Fig. 2c. The $\rho(T)$ at ambient pressure (AP) shows a high $T_c^{onset} \approx 18$ K and $T_c^{90\%Rn} \approx 17.3$ K. These values are a little bit higher than those in the previous report[44], which may attribute to the different reduction conditions. For sample No. 1 at AP, a small residual resistance below $T_c^{offset} \approx 12.2$ K is noted, which should be ascribed to the inhomogeneity of the small-sized sample for high-pressure measurements (typical size of 0.6 mm × 0.3 mm). Upon application of high pressure, we observed perfect zero resistivity for sample No. 1 below 2.4 GPa and for sample No. 2 below 4.6 GPa, respectively. At 2.4 GPa, the superconducting transition of sample No. 1 increases to $T_c^{onset} \approx 22.5$ K, $T_c^{90\%Rn} \approx 20.4$ K, and $T_c^{offset} \approx 16.5$ K. Upon further increasing pressure to 6.6 GPa, although zero resistivity cannot be achieved, the superconducting transition remains relatively sharp and is further enhanced to $T_c^{onset} \approx 26$ K, and $T_c^{90\%Rn} \approx 23.5$ K. In contrast, the superconducting transition of sample No. 2 is broadened up considerably featured by a long tail with a large residual resistivity when increasing pressure from 6.8 to 12.1 GPa, Fig. 2d. As discussed below, this should be attributed to the degradation of the delicate thin-film samples in the presence of substantial stress/strain accompanying the solidification of PTM upon compression and cooling down. Fortunately, we can still monitor the onset of superconducting transition that continues to increase with pressure. The $T_c^{onset}$ and $T_c^{90\%Rn}$ are enhanced gradually from 26 K and 22.7 K at 6.8 GPa to 31 K and 27.2 K at 12.1 GPa.

To check the reproducibility of the above results, we measure three more samples (Nos. 3, 4, and 5) with thicker SrTiO$_3$ capping layer by using different liquid PTM, i.e., mineral oil, silicone oil and glycerol, respectively. All $\rho(T)$ data are shown in Supplementary Fig. 2. We confirm that the superconducting transition $T_c^{onset}$ for all samples shows positive pressure effect. However, the superconducting transition of these three samples is broadened up significantly under pressure, which can be ascribed to the thicker capping layer that induces stronger stress/strain to the Pr$_{0.82}$Sr$_{0.18}$NiO$_2$ thin films.

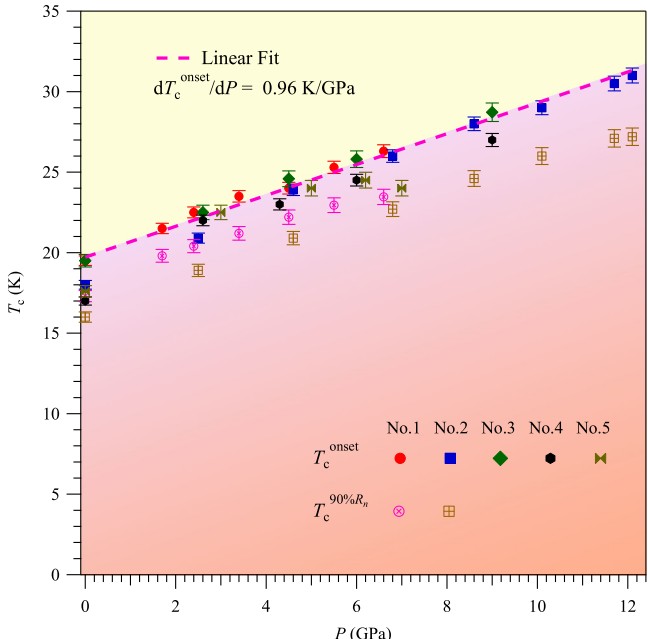

**Fig. 3 | Temperature-pressure phase diagram of Pr$_{0.82}$Sr$_{0.18}$NiO$_2$ thin films.** Pressure dependences of the superconducting transition temperatures $T_c^{onset}$ and $T_c^{90\%Rn}$ determined from the $\rho(T)$ measurements. Linear fit to the $T_c^{onset}$ gives the relation $T_c(P) = 19.7 + 0.96 \times P$. The error bars of the transition temperatures in the phase diagram are estimated from the width of the transitions.

## Temperature-pressure phase diagram

Figure 3 shows the pressure dependences of $T_c^{onset}$ and $T_c^{90\%Rn}$ obtained from the above resistivity measurements on the five samples. The positive pressure effect on $T_c$ can be visualized clearly, and the $T_c^{onset}$ is raised up monotonically from below 20 K at ambient to above 30 K at 12.1 GPa without showing any sign of saturation. This means that $T_c$ of these nickelate superconductors can be further enhanced by higher pressures, and the present study has pushed the $T_c$ limit of superconducting nickelates to over 30 K. A linear fitting to $T_c^{onset}(P)$ for all the measured samples yields a positive slope of 0.96 K/GPa, the dashed line in Fig. 3. Such an enhancement of $T_c$ will draw more attention and efforts to further boost the superconducting transition temperature of the infinite-layer nickelates superconductors.

## The upper critical field

To further characterize the superconducting properties of the Pr$_{0.82}$Sr$_{0.18}$NiO$_2$ thin films under pressure, we study the effect of magnetic fields on resistivity at each fixed pressure. Figure 4a shows the $\rho(T)$ curves under various magnetic fields at 12.1 GPa. All $\rho(T)$ data under different pressures for sample No. 1 up to 6.6 GPa and for sample No. 2 up to 12.1 GPa are shown in Supplementary Fig. 3 and Supplementary Fig. 4, respectively. As seen in Fig. 4a, Supplementary Fig. 3 and Supplementary Fig. 4, the superconducting transition is gradually suppressed to lower temperatures and the transition width is broaden up with increasing magnetic fields. Here, we used the criteria of $T_c^{90\%Rn}$ and plotted the temperature dependences of $\mu_0H_{c2}(T_c^{90\%Rn})$ in Fig. 4b. Then, we first estimate the zero-temperature upper critical field $\mu_0H_{c2}(0)$ by employing the empirical Ginzburg–Landau (G-L) formula, i.e., $\mu_0H_{c2}(T) = \mu_0H_{c2}(0)(1 - t^2)/(1 + t^2)$, where $t = T/T_c$ represents the reduced temperature. The fitting results are indicated by the broken lines in Fig. 4b. Unlike the monotonic enhancement of $T_c(P)$, the $\mu_0H_{c2}^{GL}(0)$ exhibits a non-monotonic evolution with pressure. As show in Fig. 4c, the obtained $\mu_0H_{c2}^{GL}(0)$ first increases quickly from ~100.7 T at 0 GPa to 141.5 T at 2.5 GPa with a slope of about ~16 T/GPa, and then it increases nearly linearly to ~173 T with

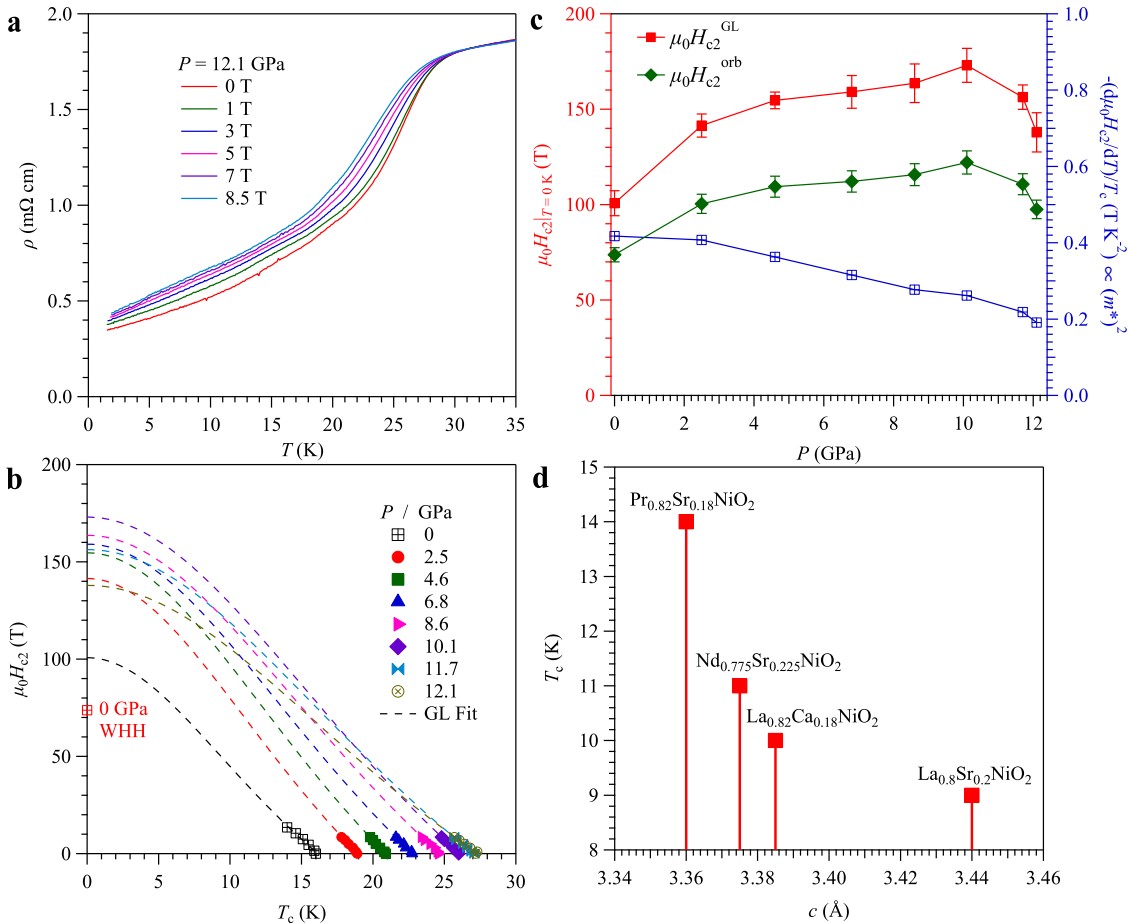

**Fig. 4 | Magneto-transport properties of the infinite-layer $Pr_{0.82}Sr_{0.18}NiO_2$ thin films and superconducting properties in nickelate thin films.** **a** Temperature dependence of the resistivity with magnetic fields up to 8.5 T at 12.1 GPa. **b** Temperature dependences of the upper critical field $\mu_0H_{c2}$ at different pressures, where $\mu_0H_{c2}$ values are determined by using the criteria of 90% $\rho_n$. The broken lines represent the Ginzburg–Landau (G-L) fitting curves and the $\mu_0H_{c2}^{orb}(0)$ calculated from WHH model at 0 GPa is indicated. **c** Pressure dependence of the zero-temperature upper critical field $\mu_0H_{c2}(0)$ obtained from the empirical G-L fitting and the WHH model, and the normalized slope $-(1/T_c)[d\mu_0H_{c2}/dT]|_{Tc}$. **d** $c$-axis dependence of $T_c$ in the series of infinite nickelates superconducting thin films. The error bars of the upper critical field in **c** are obtained from the fitting results.

a smaller slope of 4 T/GPa up to 10.1 GPa. When $P \geq 10.1$ GPa, the $\mu_0H_{c2}(0)$ declines continuously, which might be correlated with the degradation of the superconducting state. Since the temperature range in the above fitting is quite limited, the empirical G-L fit will bring quite large uncertainty for the obtained $\mu_0H_{c2}(0)$ in Fig. 4b. To overcome this problem, we also estimate the orbital-limiting $\mu_0H_{c2}^{orb}(0) = 0.69T_c \, | \, dH_{c2}/dT \, | \, _{Tc}$ in the dirty limit based on the Werthamer-Helfand-Hohenberg (WHH) model from the initial slope of $\mu_0H_{c2}(T)$ at $T_c$[59]. The calculated $\mu_0H_{c2}^{orb}(0)$ values shown in Fig. 4c are indeed smaller than those extracted from the empirical G-L fitting, but they share the similar evolution as a function of pressure.

As seen in Fig. 4a and Supplementary Fig. 4, at pressures higher than 5 GPa when the liquid PTM solidifies, the superconducting transition at zero field becomes quite broad and ended with a substantial residual resistivity that decreases continuously upon cooling and shows a positive magnetoresistance. This observation indicates that the superconducting regions are coexisting with some non-superconducting metallic regions. With increasing magnetic field, the gradual reduction of superconducting region and a positive magnetoresistance of those non-superconducting metallic region can explain the observed enhancement of resistivity at low temperatures. This effect becomes stronger as pressure increases when the non-superconducting region becomes larger.

## Discussion

The main finding of the present study is the observation of positive pressure effect on $T_c(P)$, which shows the potential to reach higher than 30 K in the $Pr_{0.82}Sr_{0.18}NiO_2$ thin films. Below we briefly discuss the mechanism and its implications.

First, the contraction of the lattice parameter under pressure should favor a higher $T_c$ when comparing to the effect of chemical pressure at similar optimal hole doping level. For the reported nickelate thin films, the in-plane lattice constants $a$ ~3.91 Å seems to be locked by the $SrTiO_3$ substrate[14,42–44], but the $c$-axis constant shows an inverse correlation to the optimal $T_c$, as illustrated in Fig. 4d, $T_c = 9$ K for $La_{0.8}Sr_{0.2}NiO_2$ ($c \approx 3.44$ Å)[42], $T_c = 10$ K for $La_{0.82}Ca_{0.18}NiO_2$ ($c \approx 3.385$ Å)[43], $T_c = 11$ K for $Nd_{0.775}Sr_{0.225}NiO_2$ ($c \approx 3.375$ Å)[14,16,17], and $T_c = 14$ K for $Pr_{0.82}Sr_{0.18}NiO_2$ ($c \approx 3.36$ Å)[44]. If this trend is followed under physical pressure, the contraction of $c$-axis under pressure should lead to an enhancement of $T_c$, as indeed observed here from ~17 K at AP to ~31 K at 12.1 GPa. From a linear extrapolation in Fig. 3, a higher $T_c$ over ~40 K can be achieved at about 20 GPa. The shrinkage of $c$-axis under pressure will enhance the hybridization between the $3d$ orbitals of nickel and $5d$ orbitals of rare-earth-layer, and thus the Kondo coupling. Following the Kondo picture, the enhancement of normal-state resistivity under high pressures can be interpreted as the magnification of enhanced Kondo effect[39,40]. Although the solidification of liquid PTM at pressures higher than 5 GPa would unavoidably introduce some defects on the delicate thin-film samples, the

application of high pressure below 5 GPa actually improves the transport property as indicated by the sharp superconducting transition with perfect zero-resistance state, *e.g.* see the $\rho(T)$ of sample No. 1 at 1.7 and 2.4 GPa in Fig. 2c and that of sample No. 2 at 2.5 and 4.6 GPa in Fig. 2d. In these low-pressure ranges, we find that both the normal-state resistivity and upper critical field indeed increase with pressure, which should be regarded as an intrinsic response to pressure. A superconducting phase diagram based on the generalized *K-t-J* model has been established by incorporating the Kondo coupling *K* to the *t-J* model, and an evolution from *d*-wave dominant phase to *s*-wave dominant phase can be achieved by tunning the coupling parameter *K* at the optimal hole-doping[40]. Therefore, the enhancement of $T_c$ may originate from this proposed picture that the gap magnitude of *s*-wave coming from the hybridized orbitals increases under high pressures.

Secondly, based on the single-band model, the initial slope of $\mu_0 H_{c2}(T)$ is related to the effective mass $m^*$ of charge carriers via the relationship of $-(1/T_c)[d\mu_0 H_{c2}/dT]|_{T_c} \propto (m^*)^2$ [60,61]. From a linear fitting to $\mu_0 H_{c2}(T)$, we can extract the normalized slope $-(1/T_c)[d\mu_0 H_{c2}/dT]|_{T_c}$, which decreases from -0.42 T/K$^2$ at 0 GPa to -0.26 T/K$^2$ at 10.1 GPa, Fig. 4c. Such a pressure-induced reduction of effective mass $m^*$ or the electron correlations should correlate with the changes of bandwidth under high pressures. As indicated by the theoretical calculations based on the one-band Hubbard model, the superconducting $T_c$ can be further enhanced by fine tuning the second- and third-nearest neighbor hopping parameters $t'$ and $t''$ on a square lattice other than hole-doping[32]. In this sense, our high-pressure results are consistent with the theoretical predictions that high pressure can create the compressive strain which broadens the bandwidth with a slight interaction-to-bandwidth ratio[32,62]. On the other hand, a continuous increase of $T_c$ from ~20 to ~35 K is observed in calculations by reducing the onsite interaction $U$ from 9$t$ to 7$t$[32], which is in good agreement with our experimental results. Further theoretical studies on the electronic structures under pressure are needed to have a better understanding on the importance of electronic correlations, multiband effects, and hybridization between Ni-3$d$ and rare-earth 5$d$ electrons.

Finally, we would like to briefly comment the influence of pressure environment on the superconducting nickelate thin films, which are found to be very sensitive to the pressure conditions or the used PTM during this study. To confirm our assumption, we perform the STEM measurements on the $Pr_{0.82}Sr_{0.18}NiO_2$ thin film (No. 2) after decompression. As shown in Supplementary Fig. 5a, we can clearly see that the atomic-resolution image of infinite-layer structure of $Pr_{0.82}Sr_{0.18}NiO_2$ thin film before we perform the high-pressure resistivity measurements. However, the infinite-layer structure is partially destroyed, producing considerable amount of dislocations and defects in a large scale after the high-pressure measurements, Supplementary Fig. 5b–d. To further confirm the existence of infinite-layer phase, we perform the synchrotron XRD measurements after the high-pressure resistance measurements. As shown in Supplementary Fig. 5e, our results confirm directly that the infinite-layer phase is preserved after high-pressure measurements, even though the characteristic 002 peak is considerably broadened up, which is accordance with the TEM results. Obviously, the broaden of the superconducting transition at high pressures has direct correlation with the cracked infinite-layer structure. Moreover, this can explain why the bulk materials don't show the superconducting transition at ambient pressure and high pressures. For bulk polycrystalline samples, although they have infinite-layer structure, there is no ideal infinite-layer structure in a large-scale as seen in the thin films. To further verify the above hypothesis about the influence of extrinsic disorders and/or stress, we perform comparative high-pressure resistivity measurements on the $Pr_{0.82}Sr_{0.18}NiO_2$ thin film (No. 6) in CAC by using the solid h-BN as the PTM, which is expected to produce a stronger stress/strain than the liquid one. As shown in Supplementary Fig. 6, the $\rho(T)$ is

immediately altered to an insulating-like behavior under pressure of 2 GPa, and the magnitude of resistivity increases significantly with further increasing pressure to 4 GPa, which reproduces the insulating behavior of the bulk polycrystalline samples. This comparison also highlights that the liquid PTM, though solidified under pressure, remains relatively soft so that it can preserve the metallic behavior and allows us to see the evolution of the superconducting transition.

In summary, we performed a high-pressure study on the superconducting $Pr_{0.82}Sr_{0.18}NiO_2$ thin films by employing the cubic anvil cell apparatus. Our results reveal that its $T_c$ increases from ~17 K at 0 GPa to ~31 K at 12.1 GPa without showing any signature of saturation. This result indicates that there is still much room for further raising the $T_c$ of the superconducting nickelates. We discuss the positive pressure effect of $T_c(P)$ in terms of the lattice contraction, enhanced hybridization between the Ni-3$d$ and Pr-5$d$ orbitals and reduced electronic correlation in light of the existing theoretical models. This finding is encouraging and should promote more studies to explore superconducting nickelates with higher $T_c$.

## Methods

### Film growth

The (001)-oriented $SrTiO_3$ (STO) substrates of the size $5 \times 5$ mm$^2$ were etched in HF solution for 40 s and then annealed at 1050 °C for 90 min with the heating and cooling rate of 10 °C/min. The 12 nm-thick $Pr_{0.82}Sr_{0.18}NiO_3$ films were deposited on the STO substrates at 650 °C with 150 mTorr oxygen partial pressure by using a pulsed laser deposition system. The 1.2 J/cm$^2$ laser fluence with 4 Hz frequency repetition was provided by a XeCl laser with 308 nm central wavelength. Subsequently, the 2 nm-thick STO film was deposited on the $Pr_{0.82}Sr_{0.18}NiO_3$ film as the capping layer under the same depositing conditions. After finishing the growth process, the precursor thin films were cooled down to room temperature at a rate of 20 °C/min. The targets were synthesized by sintering stoichiometric compounds of $Pr_2O_3$, $SrCO_3$ and NiO powder for 15 h at 1250 °C, 1300 °C and 1350 °C, respectively.

### Reduction process

Each precursor thin film was cut into four pieces of the size $2.5 \times 2.5$ mm$^2$ and then wrapped in the aluminum foil one by one. Each piece was vacuum-sealed in the glass tube (vacuum of $4 \times 10^{-4}$ torr) with different $CaH_2$ amount. And the glass tube was heated to 300 °C with the rate of 10 °C/min and kept for various reduction time. Finally, the glass tubes were cooled down to room temperature at 10 °C/min.

### X-ray characterization

The X-ray diffraction $\theta-2\theta$ symmetric scans and the reciprocal space mappings of the $Pr_{0.82}Sr_{0.18}NiO_3$ and the $Pr_{0.82}Sr_{0.18}NiO_2$ films were obtained by a Rigaku SmartLab (8 kW) high-resolution x-ray diffractometer with the wavelength of the x-ray is 0.154 nm.

### Physical properties measurement

The resistivity was measured in the commercial Physical Property Measurement System (PPMS, Quantum Design Inc.) by using the standard four-probe methods at ambient pressure.

### Scanning transmission electron microscopy

The atomic structures of these heterostructures were characterized using an ARM-200CF transmission electron microscope operated at 200 keV and equipped with double spherical aberration (Cs) correctors.

### High-pressure measurements in cubic anvil cell apparatus

Standard four-probe method was used to measure temperature-dependent resistivity of the $Pr_{0.82}Sr_{0.18}NiO_2$ thin films with the electric current applied within the *ab*-plane. The magnetic field is applied

nearly parallel to the $c$-axis. We employ a palm-type CAC to measure its resistivity under various pressures up to 12.1 GPa with the glycerol as the liquid PTM, and to 6.6 GPa with the Daphne 7373 as the liquid PTM. For comparison, we also performed similar measurements with liquid PTM mineral oil and silicone oil, and solid PTM h-BN. The pressure values inside the CAC were estimated from the pressure-loading force calibration curve pre-determined by measuring the $T_c$ of Pb at low temperatures.

## Data availability
The data that support the findings of this study are available from the corresponding authors upon reasonable request.

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

## Acknowledgements

This work is supported by the National Key R&D Program of China (2018YFA0305700, J.G.C. and 2019YFA0308500, K.J.J.), the Beijing Natural Science Foundation (Z190008, J.G.C.), the National Natural Science Foundation of China (12025408, J.G.C.; 11904391, J.P.S.; 11921004, J.G.C.; 11888101, J.G.C.; 11834016, X.L.D.; 11874400, J.G.C.; 12174424, J.P.S.; and 11721404, K.J.J.), the Strategic Priority Research Program and Key Research Program of Frontier Sciences of the Chinese Academy of Sciences (XDB25000000, J.P.S.; XDB33000000, J.G.C.; and QYZDB-SSW-SLH013, J.G.C.), the Users with Excellence Program of Hefei Science Center CAS (Grant No. 2021HSC-UE008, J.G.C.) and the CAS Interdisciplinary Innovation Team (GJTD-2020-01, J.G.C.). Y.U. is supported by the JSPS KAKENHI (Grant No. JP19H00648, Y.U.). The synchrotron XRD measurements were performed at the 1W1A Station of the Beijing Synchrotron Radiation Facility (BSRF).

## Author contributions

J.G.C., K.J.J. and J.P.S supervised this project. M.W.Y., Z.Y., Z.H.Z and K.J.J. synthesized and characterized the $Pr_{0.82}Sr_{0.18}NiO_2$ thin films at ambient pressure. N.N.W., H.Z., J.P.S. and X.L.D. measured the physical properties of thin films at ambient pressure. N.N.W., K.Y.C. and J.P.S. performed high-pressure resistivity measurements and data analyses. Q.H.Z. and L.G. carried out the scanning transmission electron microscopy measurements. Y.U. provides the high-pressure methodology. All authors discussed the results. J.P.S. and J.G.C. wrote the paper with inputs from all authors.

## Competing interests

The authors declare no competing interests.
