## [Peer Review File · Nature Communications]

Pressure-induced monotonic enhancement of T_c to over 30 K in the superconducting $\text{Pr}_{0.82}\text{Sr}_{0.18}\text{NiO}_2$ thin filmsREVIEWER COMMENTS

Reviewer #1 (Remarks to the Author):

The hole-doped nickelate 112 thin film exhibits superconductivity with T_c up to about 18 K. There are several reasons to argue that the superconductivity mechanism has some similarities as the cuprates, both from many theoretical works and the recent tunneling measurement of the d-wave gap. It remains however to know whether the T_c can be improved further. The authors here conducted a high-pressure study on the superconducting $\text{Pr}_{0.82}\text{Sr}_{0.18}\text{NiO}_2$ thin films by employing the cubic anvil cell apparatus. They pushed the T_c higher up to about 31 K at 12.1 GPa without showing any signature of saturation. This is a challenging experiment which may only be successful when liquid transmitting materials are used. This result strongly suggests that there is still more room for improving T_c , thus researchers in this field are all expecting that. The data are clearly presented without any doubt. And the paper is well written. The discussions about the absence of superconductivity in depressed thin films and bulk samples are also reasonable. Thus I strongly recommend the acceptance of this work for publication in Nature Communications.

Reviewer #2 (Remarks to the Author):

In this manuscript, the authors report an over 10K increase of superconducting transition onset temperature in $\text{Pr}_{0.82}\text{Sr}_{0.18}\text{NiO}_2$ thin films under hydrostatic pressure of up to 12.1 GPa. This material has garnered lots of attention as it presents a new family of unconventional superconductors and also a close analogy to the high temperature cuprate superconductors. However, experimental characterization of the material's inter-layer coupling and its relevant has been lacking. It is also unclear whether the currently reported superconducting transition temperature is intrinsic to the material or limited by its thin film form. In this sense, the manuscript provides timely and interesting results from a high-pressure perspective, which would be of broad interest to the field. I recommend its publication in nature comm, if the authors can address the following remarks.

- 1) The author provides interesting discussion of the pressure environment on the film near the end. In the supplementary figure 5, the author shows TEM of the film post high-pressure, where clear lattice distortion is visible. In certain regions, lattice rotation is seen showing a tendency to possibly form c-axis in plane domains. In others, the unit cells look more cubic than tetragonal to me. Are there any oxygen stoichiometry change during the exposure to high pressure? Do the authors have XRD data of the film after high-pressure measurement showing that the 112 phase survives?
- 2) The author introduced 3 main theoretical perspectives but primarily focused on interpreting the experimental results in terms of Kondo coupling between R 5d and Ni 3d orbitals. Are the authors suggesting that the data favors this perspective? Or the other two theoretical models can also be compatible? The authors should be clearer.
- 3) The author investigated the upper critical field and resistivity evolution as function of the pressure. However, it is not clear to me the observed dependences are intrinsic as the pressure destroys the crystalline integrity of the film and turns into effectively a mixed phase. The empirical GL fit also seem to significantly overestimate the low temperature upper critical field: the fitted zero temperature H_{c2} is higher than the intersect point of a linear extrapolation of the high temperature data point to zero temperature.
- 4) Can the author comment on why in Fig. S4, the low temperature resistivity saturation develops a nontrivial field dependence as the pressure increases?

Minor comment:

- In abstract, the chemical reduction is topotactic not topological

Reviewer #3 (Remarks to the Author):

The recently reported superconductivity of 9-15 K in an infinite-layer nickelate $\text{Nd}_{0.8}\text{Sr}_{0.2}\text{NiO}_2$ thin film has caused a broad interest (Nature 572, 624 (2019)). Even though the crystalline structure and electronic configuration of infinite-layer nickelates are similar to that of high- T_c cuprates, compared to copper and iron-based superconductors, nickelates still show low T_c for all recently found materials $(\text{La}/\text{Pr}/\text{Nd})(\text{Sr}/\text{Ca})\text{NiO}_2$. This may question whether the nickelate is a potential candidate for high T_c superconductors. In this manuscript, the authors performed the pressure study on transport property of high-quality $\text{Pr}_{0.82}\text{Sr}_{0.18}\text{NiO}_2$ thin films, and found the T_c could be enhanced from 17K at ambient pressure to 31 K at 12.1 GPa. More importantly, the authors found that the T_c did not show saturation with increasing pressure up to ~ 12 GPa, suggesting that the T_c can be further enhanced if much higher pressure could be applied. The current result is an encouraging step in the field of nickelate superconductivity. In these years, pressure-induced T_c enhancements is a hot research area and have performed on various materials, e.g., on Hydride with pressure above 200 GPa and the T_c can reach room temperature. Considering that the current pressure in this study is still low (12 GPa), it may be potential to see higher T_c at higher pressure in nickelates in the future study. Overall, the result is interesting, and the analysis of data is well organized and the manuscript is well written. I recommend the publication of this work in Nature Communications.

Reply to the Report of Referees

Reviewer 1:

The hole-doped nickelate 112 thin film exhibits superconductivity with T_c up to about 18 K. There are several reasons to argue that the superconductivity mechanism has some similarities as the cuprates, both from many theoretical works and the recent tunneling measurement of the d-wave gap. It remains however to know whether the T_c can be improved further. The authors here conducted a high-pressure study on the superconducting $\text{Pr}_{0.82}\text{Sr}_{0.18}\text{NiO}_2$ thin films by employing the cubic anvil cell apparatus. They pushed the T_c higher up to about 31 K at 12.1 GPa without showing any signature of saturation. This is a challenging experiment which may only be successful when liquid transmitting materials are used. This result strongly suggests that there is still more room for improving T_c , thus researchers in this field are all expecting that. The data are clearly presented without any doubt. And the paper is well written. The discussions about the absence of superconductivity in depressed thin films and bulk samples are also reasonable. Thus, I strongly recommend the acceptance of this work for publication in Nature Communications.

Reply: Thank you so much for your careful review, high remarks and recommendation to our work.

Reviewer 2:

In this manuscript, the authors report an over 10K increase of superconducting transition onset temperature in $\text{Pr}_{0.82}\text{Sr}_{0.18}\text{NiO}_2$ thin films under hydrostatic pressure of up to 12.1 GPa. This material has garnered lots of attention as it presents a new family of unconventional superconductors and also a close analogy to the high temperature cuprate superconductors. However, experimental characterization of the material's inter-layer coupling and its relevant has been lacking. It is also unclear whether the currently reported superconducting transition temperature is intrinsic to the material or limited by its thin film form. In this sense, the manuscript provides timely and interesting results from a high-pressure perspective, which would be of broad interest to the field. I recommend its publication in nature comm, if the authors can address the following remarks.

Reply: Thank you so much for spending your valuable time to review our manuscript. We appreciated your positive remarks and recommendation to our work. We have revised the manuscript according to your suggestions as listed below.

1) The author provides interesting discussion of the pressure environment on the film near the end. In the supplementary figure 5, the author shows TEM of the film post high-pressure, where clear lattice distortion is visible. In certain regions, lattice rotation is seen showing a tendency to possibly form c-axis in plane domains. In others, the unit

cells look more cubic than tetragonal to me. Are there any oxygen stoichiometry change during the exposure to high pressure? Do the authors have XRD data of the film after high-pressure measurement showing that the 112 phase survives?

Reply: We thank the referee for pointing out this intriguing issue. Honestly, we did not evaluate the oxygen content of the superconducting film before and after high-pressure resistance measurements because such a task is very difficult considering the thin-film sample covered with some capping layer. Usually, the oxide samples are expected to preserve the oxygen stoichiometry below room temperature even in the exposure to high pressure.

In our study, we cut the bigger samples ($2.5 \times 2.5 \text{ mm}^2$) into small pieces ($0.6 \times 0.3 \text{ mm}^2$) and picked out the high-quality samples with a higher T_c for resistivity measurements under high pressures. For such a small sample, it is very difficult to collect XRD data in a laboratory XRD machine. To address your concern, we resorted to the synchrotron-based XRD measurements. Figure R1 shows the synchrotron XRD patterns of $\text{Pr}_{0.82}\text{Sr}_{0.18}\text{NiO}_2$ film before and after high-pressure resistance measurements. These results confirm directly that the 112 phase is preserved after high-pressure measurements, even though the characteristic 002 peak is considerably broadened up, which is in well accordance with the TEM results. In the revised manuscript, we have added the synchrotron XRD data in Fig. S5 of the Supplementary Materials.

Fig. R1 Synchrotron X-ray diffraction θ - 2θ symmetric scans of infinite-layer $\text{Pr}_{0.82}\text{Sr}_{0.18}\text{NiO}_2$ thin film before and after high-pressure resistance measurements with a wavelength of $\lambda = 0.6199 \text{ \AA}$.

2) The author introduced 3 main theoretical perspectives but primarily focused on interpreting the experimental results in terms of Kondo coupling between R 5d and Ni 3d orbitals. Are the authors suggesting that the data favors this perspective? Or the other two theoretical models can also be compatible? The authors should be clearer.

Reply: Thank you for the comment. In addition to the Kondo picture, in the third paragraph of Discussions, we also discussed the observed monotonic reduction of $-(1/T_c)[dH_{c2}/dT]_{T_c}$ that is proportional to the effective mass m^* in light of the one-band Hubbard model, which predicts that T_c can be enhanced by increasing the hopping energy (or reducing the electron correlations). Thus, our experimental finding of monotonic enhancement of $T_c(P)$ is consistent with the prediction of one-band Hubbard model given in Ref. 32. It should be noted this work is an experimental study and cannot distinguish existing theoretical models, especially those based on the single-orbital versus the multi-orbitals. Instead, we have attempted to discuss our experimental results in light of the available theoretical perspectives and provide some implications for further studies.

3) The author investigated the upper critical field and resistivity evolution as function of the pressure. However, it is not clear to me the observed dependences are intrinsic as the pressure destroys the crystalline integrity of the film and turns into effectively a mixed phase. The empirical GL fit also seem to significantly overestimate the low temperature upper critical field: the fitted zero temperature H_{c2} is higher than the intersect point of a linear extrapolation of the high temperature data point to zero temperature.

Reply: Thank you for the valuable suggestion. Indeed, the fragile thin-film samples undergoes some irreversible damage under pressures higher than 5 GPa when the employed liquid pressure transmitting medium solidifies, which will unavoidably influence the normal- and superconducting-state properties of the studied sample. However, at pressures lower than 5 GPa, the application of high pressure actually improves the sample quality as indicated by the sharp superconducting transition with perfect zero-resistance state, *e.g.* see the $\rho(T)$ of sample No. 1 at 1.7 and 2.4 GPa in Fig. 2(c) and the $\rho(T)$ of sample No. 2 at 2.5 and 4.6 GPa in Fig. 2(d). In these low-pressure ranges, we found that both the normal-state resistivity and upper critical field increase with pressure, which should be regarded as an intrinsic response to pressure. At higher pressure, the degradation of the thin-film samples indeed introduces some irregular variations of these quantities, but the general trends similar to that at lower pressure are retained. In the revised manuscript, we have emphasized that our discussions are mainly based on the intrinsic pressure response at lower pressure range when the sample retains a relatively good quality.

We totally agree with you that the empirical GL fit will significantly overestimate the zero-temperature upper critical field given the rather limited temperature range in Fig. 4(b). To overcome this problem, we have also estimated the orbital-limiting $\mu_0 H_{c2}^{\text{orb}}(0) = 0.69T_c[dH_{c2}/dT]_{T_c}$ in the dirty limit based on the Werthamer-Helfand-Hohenberg (WHH) model from the initial slope of $H_{c2}(T)$ at T_c . The calculated $\mu_0 H_{c2}^{\text{orb}}(0)$ values

are indeed smaller than those extracted from the empirical GL fit, but they share the similar evolution as a function of pressure, see Fig. 4(c) in the revised manuscript. Thus, our discussions are still valid. In the revised manuscript, we have added the calculated $\mu_0 H_{c2}^{\text{orb}}(0)$ data in Fig. 4(c) for a comparison.

4) Can the author comment on why in Fig. S4, the low temperature resistivity saturation develops a nontrivial field dependence as the pressure increases?

Reply: As seen in Fig. S4, at pressures higher than 5 GPa when the pressure transmitting medium solidifies, the superconducting transition at zero field becomes quite broad and ended with a substantial residual resistivity that decreases continuously upon cooling. This observation indicates that superconducting regions are coexisting with some non-superconducting metallic regions. With increasing magnetic field, the gradual reduction of superconducting region and a positive magnetoresistance of those nonsuperconducting metallic region can explain the observed enhancement of resistivity at low temperatures. This phenomenon should become stronger as pressure increases when the non-superconducting region becomes larger. We have added some discussions mentioned above in the revised manuscript.

Minor comment:

- In abstract, the chemical reduction is topotactic not topological.

Reply: Thank you for pointing out this mistake, which is corrected in the revised manuscript.

Reviewer 3:

The recently reported superconductivity of 9-15 K in an infinite-layer nickelate $\text{Nd}_{0.8}\text{Sr}_{0.2}\text{NiO}_2$ thin film has caused a broad interest (Nature 572, 624 (2019)). Even though the crystalline structure and electronic configuration of infinite-layer nickelates are similar to that of high- T_c cuprates, compared to copper and iron-based superconductors, nickelates still show low T_c for all recently found materials (La/Pr/Nd)(Sr/Ca)NiO₂. This may question whether the nickelate is a potential candidate for high T_c superconductors. In this manuscript, the authors performed the pressure study on transport property of high-quality $\text{Pr}_{0.82}\text{Sr}_{0.18}\text{NiO}_2$ thin films, and found the T_c could be enhanced from 17 K at ambient pressure to 31 K at 12.1 GPa. More importantly, the authors found that the T_c did not show saturation with increasing pressure up to ~12 GPa, suggesting that the T_c can be further enhanced if much higher pressure could be applied. The current result is an encouraging step in the field of nickelate superconductivity. In these years, pressure-induced T_c enhancements is a hot research area and have performed on various materials, e.g., on Hydride with pressure above 200 GPa and the T_c can reach room temperature. Considering that the current pressure in this study is still low (12 GPa), it may be potential to see higher T_c at higher pressure in nickelates in the future study. Overall, the result is interesting, and the analysis of data is well organized, and the manuscript is well written. I recommend the publication of this work in Nature Communications.

Reply: Thank you so much for your careful review, high remarks and recommendation

to our work.

REVIEWERS' COMMENTS

Reviewer #2 (Remarks to the Author):

The revised manuscript has provided satisfactory answers to the questions I raised in my previous review. This work has shown that there is still substantial room for T_c improvement in nickelate superconductors. I recommend this manuscript for publication now.

Dear Dr. Paul Wiecki,

Thank you for handling our manuscript and informing us the good news to publish a suitably revised version in Nature Communications. In this round, the referee raised no further concerns and recommended our manuscript for publication. Thus, we mainly revised the manuscript carefully to comply with the policies and formatting requirements of Nature Communications. We provided high-quality version of each figure in the main text. Also, we prepared the image of “Temperature-pressure phase diagram of $\text{Pr}_{0.82}\text{Sr}_{0.18}\text{NiO}_2$ thin film” for your consideration as a Featured Image on the Nature Communication homepage. We hope that the revision of our manuscript could meet the requirement for the final publication. Please contact me if any information is further needed.

Yours Sincerely;

Jianping Sun

On behalf of all coauthors